# Oral Health Related Quality of Life and Prosthetic Status among Institutionalized Elderly from the Bucharest Area: A Pilot Study

**DOI:** 10.3390/ijerph18126663

**Published:** 2021-06-21

**Authors:** Laura Iosif, Cristina Teodora Preoteasa, Elena Preoteasa, Ana Ispas, Radu Ilinca, Cǎtǎlina Murariu-Mǎgureanu, Oana Elena Amza

**Affiliations:** 1Department of Prosthodontics, Faculty of Dental Medicine, “Carol Davila” University of Medicine and Pharmacy, 17-21 Calea Plevnei Street, Sector 5, 010221 Bucharest, Romania; laura.iosif@umfcd.ro (L.I.); dr_elena_preoteasa@yahoo.com (E.P.); dr_catalina_magureanu@petralaboratory.ro (C.M.-M.); 2Department of Ergonomics and Scientific Research Methodology, Faculty of Dental Medicine, “Carol Davila” University of Medicine and Pharmacy, 17-21 Calea Plevnei Street, Sector 5, 010221 Bucharest, Romania; 3Department of Prosthodontics, Faculty of Dental Medicine, “Iuliu Hatieganu” University of Medicine and Pharmacy, 32 Clinicilor Street, 400006 Cluj-Napoca, Romania; 4Department of Biophysics, Faculty of Dental Medicine, “Carol Davila” University of Medicine and Pharmacy, 17-21 Calea Plevnei Street, Sector 5, 010221 Bucharest, Romania; radu.ilinca@umfcd.ro; 5Department of Endodontics, Faculty of Dental Medicine, “Carol Davila” University of Medicine and Pharmacy, 17-21 Calea Plevnei Street, Sector 5, 010221 Bucharest, Romania; oana.amza@umfcd.ro

**Keywords:** quality of life, care home, elderly, edentulism, prosthodontic status, public health

## Abstract

The aim of the study was to assess the oral health related quality of life (OHRQoL) of elderly in care homes, one of Romania’s most vulnerable social categories, to correlate it to sociodemographic, oral health parameters, and prosthodontic status. Therefore, a cross-sectional study was performed on 58 geriatrics divided into 3 age groups, who were clinically examined and answered the oral health impact profile (OHIP-14) questionnaire. Very high rates of complete edentulism in the oldest-old subgroup (bimaxillary in 64.3%; mandibular in 64.3%; maxillary in 85.7%), and alarming frequencies in the other subgroups (middle-old and youngest-old), statistically significant differences between age groups being determined. The OHIP-14 mean score was 14.5. Although not statistically significant, females had higher OHIP-14 scores, also middle-old with single maxillary arch, single mandibular arch, and bimaxillary complete edentulism, whether they wore dentures or not, but especially those without dental prosthetic treatment in the maxilla. A worse OHRQoL was also observed in wearers of bimaxillary complete dentures, in correlation with periodontal disease-related edentulism, in those with tertiary education degree, and those who came from rural areas. There were no statistically significant correlations of OHRQoL with age, total number of edentulous spaces or edentulous spaces with no prosthetic treatment. In conclusion, despite poor oral health and prosthetic status of the institutionalized elderly around Bucharest, the impact on their wellbeing is comparatively moderate.

## 1. Introduction

In the last century, life expectancy in Romania has doubled, from an average of 36.4 in 1910 to 75.5 in 2019 [1]. The increase in life expectancy globally, as well as in Romania, in recent decades [2] has multiple socio-economic implications [3]. Medical needs and requirements for ensuring a high standard and accessibility of health care for the whole society are increasing, but the Romanian medical system finds itself in a permanent crisis, a major “pain” [4] of our society after the fall of communism in Eastern Europe in 1989. Access to and cost coverage of dental services still prove difficult for the vulnerable elderly [5,6], considering the fact that Romania is a country belonging to European Union where dentistry is mainly private. Thus, adult patients in our country, including geriatric individuals, must pay all of the total cost of dental care, except the dental emergencies, mainly tooth extractions and endodontic drainages, the public health services excluding almost entirely the coverage of restorative prosthetic treatments. Institutionalized elderly encounter further barriers such as the loss of autonomy and a lack of dental units in most of the care homes in our country. 

An important aspect related to the oral health of the elderly is tooth loss, with a higher prevalence of extensive and complete edentations in elderly people compared to all other age categories [7]. Resulting changes such as, sometimes, dramatic alterations in facial appearance, the decrease in maxillary and mandibular bone volume causing problems for prostheses, the deterioration of the nutritional and implicitly general status [8], but also accentuated psycho-social changes, have a negative impact on their quality of life. As a result of various diseases that can cause tooth loss, depending on the number of edentulous spaces, and on the aggravation and complication of pre-existing prosthetic treatments [9], the oral pathology of the institutionalized elderly may include the most complex situations of partial, subtotal, and complete edentation, with difficult therapeutic perspectives.

In the elderly population, senescence can be only determined chronologically, and many studies [10,11,12,13,14,15] classify elderly adults aged 65–74 years as youngest-old, 75–84 years as middle-old, and those over 85 years as oldest-old, as proposed at the Canadian symposium of Geriatrics in 1998. This staging reflects the progression of pathophysiological changes in all systems and apparatuses, including the dentomaxillary, with the various stages of senescence. Thus, the aim of our study was to asses edentulism, dental prosthetic status, and the oral health related quality of life (OHRQoL) related to age group according to the aforementioned classification and other factors in elderly institutionalized living in long-care centers in the area of Bucharest, the capital of Romania, in 2021. The main hypothesis was that of decreasing of OHRQoL as the geriatrics progress from one age group to another.

## 2. Materials and Methods

A cross-sectional pilot study on a sample of institutionalized elderly was conducted at the beginning of May 2021. The study protocol was approved by the Ethics Commission of the Scientific Research of the Carol Davila University of Medicine and Pharmacy in Bucharest, Romania, with the corresponding ethical approval no. 11388/7.05.2021. Data were collected from 3 privately funded care homes in the vicinity of Bucharest, which allowed us, from the 8 initially asked, to interview and to examine the residents in their long care. Regarding the dental care services for those geriatrics, it has to be mentioned that none of the 3 care homes supported internal oral treatments by establishing suitable facilities for dentists, such as providing mobile dental units or treatment rooms located at the residential homes, all of them mentioning the transportation of residents to a dental office, performed mainly in cases of dental emergencies. 

A written informed consent was made available to the subjects, at the same time they received detailed verbally clarifications on the duration, purpose, and the manner of conducting the study. Inclusion criteria were the minimum age of 65 years and at least one edentulous space, which was no treated or treated with either a fixed partial denture (FPD), a removable partial denture (RPD), or a complete denture at the level of the entire madibular/maxilary arch. Exclusion criteria were the mixed prosthodontic treatment (e.g., FPD’s and RPD’s) on the same arch, neuro-mental disorders (e.g., Alzheimer’s disease, schizophrenia) and any mental or physical conditions that would have made data collection impossible. Thus, of the 175 elderly institutionalized in the long-stay centers previously mentioned at the time of the study, 152 subjects agreed to participate and of these, only 66 had met the eligibility criteria. No incentives were used for study participation.

The data were collected through interviews and clinical examinations performed by an experienced dentist belonging to the research group, wearing a face shield and using disposable medical gowns, consultation kits, and gloves. Most of the subjects were examined sitting in chairs under natural illumination. Torch light was used when required. The subjects were first interviewed to obtain sociodemographic data: age, gender, origin (rural/urban areas), education, etiology of tooth loss, and data for the assessment of OHRQoL. The type of prosthesis was clinically recorded by dentist for each participant.

OHRQoL was assessed using the Romanian version of the previously validated and reliable OHIP-14 questionnaire [16], a simplified form of the OHIP-49 questionnaire, designed by Gary D. Slade with good reliability, validity, and precision [17], and introduced in English in the U.S.A., subsequently translated and validated in other countries, such as Spain [18], Greece [19], Vietnam [20], China [21] Scotland [22] Korea [23], Portugal [24], Germany [25], Japan [26] Brazil [27], etc.

The subjects filled out the OHIP-14 form individually. Each of OHIP-14 items has a set of possible answers distributed in a 5-point Likert scale and are representative for various categories, such as functional limitations, physical pain, psychological discomfort, physical, psychological disabilities, and social disabilities and handicaps [28]. The points associated with each answer were added for each subject, resulting in a total score with possible values between 0 and 56. A higher score meant a worse OHRQoL, while a lower score indicated a better OHRQoL.

For the oral status, we recorded complete for the maxilla and the mandible, and for the latter subtotal edentations (1–2 remaining teeth) or edentations according to the Kennedy (K) classification [29], the total number of edentulous spaces, and the number of edentulous spaces without prosthetic rehabilitation. For the prosthetic status, we recorded the type of prostheses used (complete dentures; removable partial dentures—RPD; fixed partial dentures—FPD) or if edentulism was untreated. 

Statistical analysis was conducted by using the SPSS Statistics 20.0 software (IBM Corp., Armonk, NY, USA). For group comparison Fisher’s exact test, chi-square test, Mann–Whitney test or Kruskal-Wallis test were used accordingly to data type and distribution. The non-parametric Spearman test was used for correlation analysis. *p*-values less than 0.05 were considered statistically significant.

## 3. Results

The final study sample included 58 institutionalized elderly, 8 cases being withdrawn from the analysis, because of incomplete or incorrect responses to the questionnaire. The study group was aged 65–103 years (median 81.5 years), 16 participants in the youngest-old category, 28 participants in the middle-old category, and 14 participants in the oldest-old-category. Women represented the vast majority (*n* = 46) in all age groups, with the highest prevalence in the oldest-old group. The level of education was relatively similar within the groups. The majority of the youngest-old came from rural, the oldest-old came from urban areas (Table 1).

The main cause for the losing teeth were dental caries which were met at the level of the youngest-old (*n* = 12; 75%), and mixed etiology (caries and periodontal disease) was encountered in the two older categories (*n* = 20; 71.4% of middle-old; *n* = 9; 64.3% of oldest-old).

Bimaxillary complete edentation was observed in over half of the oldest-old (64.3%, *n* = 9), in about 1/3 of the middle-old (35.7%, *n* = 10), and was almost absent in the youngest-old (6.2%, *n* = 1) (Table 2). Maxillary complete edentation was encountered with a much higher frequency compared to mandibular edentation in the oldest-old group (85.7% vs. 64.3%), while in the middle-old group the two had a similar prevalence, and in the youngest-old group mandibular edentation was more common. The difference between the prevalence of complete edentation and the partial edentation in the age groups was statistically significant for the maxilla (*p* < 0.001), but not for the mandible (*p* = 0.103). Edentations without prosthetic treatment in the maxilla were more frequent in youngest-old, and in the mandible in the oldest-old and the youngest-old. Regarding the maxillary prosthetic rehabilitation (Table 2), the vast majority of those in the oldest-old group were complete denture wearers, the most of the removable partial denture wearers belonged to the middle-old category, while, the most of the youngest-old elderly were FPD’s wearers. At the mandibular level, complete and removable denture wearers showed a similar distribution in all age categories, at the same time, most of the FPD wearers being from the youngest-old category.

Regarding OHRQoL, OHIP-14 score ranged from 0 to 48 (median 14.5). Females, middle-old with single maxillary, single mandibular and bimaxillary complete edentulism, whether they wore dentures or not, but especially those without prosthetic treatments in the maxilla, recorded higher scores of OHIP-14 (Table 3), meaning worse OHRQoL. In relation to prosthetic status, worse OHRQoL was observed in those with bimaxillary complete dentures, followed by those with complete dentures in the mandible.

Although subgroup analysis showed that OHRQoL was statistically similar in treated and untreated mandibular completely edentulous patients (*p* = 0.338), higher OHIP-14 scores were registered in the mandibular denture wearer subgroup, and, when periodontal disease was found to be the main etiological factor for edentulism, also in those with tertiary education degree, and those from rural areas. There were no statistically significant correlations of OHIP with age, total number of edentulous spaces, and edentulous spaces with no prosthetic treatment, respectively.

## 4. Discussions

Institutionalized elderly people have particular and significant needs compared to the independent geriatric population and face greater barriers to receiving dental care, their oral health being generally poor [30].

In view of the scarcity of studies and social and health programmes among this population segment of our country, the present study consisted in objectively and thoroughly documenting the oral and prosthetic status of these elderly. In our study, the middle-old represented the largest age category, data which concurs with the findings of another recent study from Galati, Romania [6]. This can be explained by the increase in life expectancy in Romania, from 71.2 years to 75.5 years in the last 10 years, according to the report published by the European Commission on 28 November 2019 and by the higher autonomy and better general health status, and, therefore, the lesser need for institutionalization among the youngest-olds. Although we expected the perception of the OHRQoL to worsen from one age subgroup to another, we did not find a significant correlation between OHIP-14 scores and age, the results being similar to those of recent reports [31]. Furthermore, despite the evidence that oral health worsens with age [32] OHIP-14 values in our study, analysed by age subgroup, are even lower than those reported by a 2015 study [33], possibly because of our citizen’s low expectations based on a widespread cultural attitude of resignation and high tolerance towards oral health and the quality of prosthetic restorations.

In line with various specialized studies, some signaling, alongside the major gender disparity, a survival rate of women compared to men of over 7 years [34], most participants in our study were females (85.7% in the oldest-old category). These felt more severely affected in terms of OHRQoL than the men in our study. Studies with similar results [35,36,37] attributed this correlation to the higher sensitivity of female emotions and women’s social problems. Similarly, other findings revealed that females express, under comparable conditions, more dissatisfaction with their appearance and more complaints about pain and the inability to chew than males [38].

There are no data on the education level of Romania’s institutionalized elderly, and only limited studies that report on tooth loss or OHRQoL in relation to the education level for this population segment [39]. We recorded a higher level of dissatisfaction regarding the OHRQoL among those with higher education, which could be explained by increased sanogenic exigencies among this category. We found divergent trends in previous research, a study from the adult dental health survey (ADHS) developed in the United Kingdom in 2016 supporting our results, but not explaining the obtained values [40]. Another earlier report, also from the UK, London, shows a decrease in OHIP-14 scores with an increase in the level of education, the authors motivating their results with a higher level of culture and better oral health care among subjects with tertiary education degree [41].

The environment of origin criterion revealed in our study, an inhomogeneous group in terms of urban and rural distribution. National and international research does not provide comparative data. The fact that most of the youngest-old came from urban areas, while the older ones from rural areas—even if the differences between the groups were minor—could be explained by the lower financial possibility of those in rural areas to early access to institutionalized care. A lower OHRQoL-level among the elderly from rural areas, suggests the access to a lower quality of dental care (prior to institutionalization), worse living conditions in rural communities, and difficulties in obtaining appropriate prosthetic treatments, once they become institutionalized. Additionally, the result indicates a lower quality of life among institutionalized geriatricians in Romania from rural areas, not least from the perspective of increased oral morbidity compared to urban areas, thus representing a first report in the literature in this regard, as far as we are currently aware of.

Edentulism is an indicator of the oral health of a population and elderly people generally have a more affected oral status than younger people. The institutionalized elderly in our study presented a rate of complete edentation even double that of some specialized data which evaluated similar groups [42], although the frequency of partial edentation within the same group was clearly higher. Among the many oral diseases and disabilities elderly people in Romania suffer from complete loss of teeth in the oldest-old with a frequency of over 60% in 2020, regardless of location (upper or lower jaw), is a major health concern indicator. Recent reports that indicate a similar frequency among institutionalized elderly in Brazil, of 60% [43] are worth mentioning, but they are not able to reduce the negative impact on the OHRQoL of the completely edentulous elderly from our group. 

In relation to age subgroups, our study shows a tendency to earlier occurrence of complete mandibular than of complete maxillary edentation. This trend may explain, given the difficulty to provide stable dentures in flat resorbed mandibular ridges [44], the much poorer quality of life perception among those oldest-old wearers of total mandibular prostheses, with a long history of edentation, compared to those without protheses. In turn, the absence of prosthetic rehabilitation with complete dentures in the maxilla, most likely stemming from mainly aesthetic expectations, associates the worse OHRQoL values, the vast majority of the participants in this study being female.

In our study, the youngest-old presented mainly edentulism suitable for fixed prosthetic treatments such as fixed partial dentures (FPD), while in the middle-old and oldest-old subgroups terminal edentations made up the majority of cases, removable partial dentures (RPD) thus gaining increased eligibility. Similar findings from 2018 [45] show an increase in Kennedy Class I and Class II edentations associated with the progression in age, the explanation being, as it is in the case of our study, that the older people are, the more teeth are extracted due to multiple causes. In relation to prosthetic rehabilitation, wearers of partial dentures had worse OHRQoL values, compared to fixed dentures wearers (for the maxillary as well as for the mandible), consistent with data from an older study [46], which showed that OHRQoL differed substantially in FPD’s and RPD’s wearers, even in the case of complete denture wearers.

Edentulous spaces with no prosthetic treatment represented almost half of the total edentulous spaces in our study, the probable explanation being the underuse of dental treatments among institutionalized older people due to financial constraints, reduced mobility, lack of dental facilities in care homes [47], severe chronic diseases, and psychological factors. Untreated edentations were more prevalent in the maxilla in the youngest-old, and more prevalent in the mandible in the oldest-old, suggesting a change in the need of prosthetic treatment with age. Contrary to our results, other recent data [48] suggested a slightly higher need of prosthetic treatment of the maxilla compared to the mandible, but the study did not consider an evaluation according to age subgroups. We compared the OHRQoL of the elderly with untreated complete edentations in the maxilla and in the mandible to the average value for the whole group and to values obtained in other studies. The OHIP-14 values of 19 and 17 for those with untreated complete edentations in the maxilla and in the mandible, respectively, although only slightly higher than the average for our group are almost double compared to an overall score (regardless of the location of the complete edentation) from a recent study [49], and even 5 times higher compared to another [50], both studies being conducted on groups with similar socio-demographic characteristics, but with a much lower frequency of untreated complete edentations. This result shows that, in our study, untreated complete edentulousness (both uni- and bimaxillary) affects the perception of the quality of life of the institutionalized elderly the most, due to functional, aesthetic or psychological changes, an aspect all the more serious, as this form of edentation is the most prevalent in this group.

Periodontal disease, an important etiological factor for tooth loss, causes, in severe stages, bleeding gums, mobile teeth, pain on chewing, halitosis, and aesthetic problems, the dramatic post-extractional resorption of the edentulous ridge, especially in the elderly, posing major problems to any attempt at prosthetic rehabilitation of profoundly disabled oral functions. In the same population category, periodontitis is associated with certain systemic diseases, as established by the consensus report of the joint EFP/AAP Workshop in 2013 [51], the association with poor OHRQL in elderly being also cited [52,53]. These data raise the question of whether people with periodontal disease as the main etiological factor for tooth loss present a worse OHRQL compared to those with different edentulism causes. Indeed, our study revealed the worst OHRQL-level among those who reported tooth loss as due to periodontal disease, most probably due to ridge pain and difficulties performing oral functions, associated with discomfort in wearing dentures. These findings are, to the best of our knowledge, the first scientific report of this kind.

Finally, we must consider some of the limitations of the study, such as the low sample size which limits the generalization of the findings, given the difficulty of enrolling a larger number of participants in this population category, and possible memory bias in the recollection of the main cause of tooth loss. Severe problems, such as major depression, or manual dexterity, complicated our research recruitment. Therefore, avoiding such barriers regarding the exclusion criteria by minimizing them due including more persons after appropriate management of such conditions [54], may contribute to improve both the study design and the limited generalizability of our sample results. Further, difference categorization, e.g., structuring two groups instead of three groups used in this study, it is the one of the ways to confirm the observed trends and to obtain strong generalizable conclusions.

Other difficulties encountered in cooperating with the long care institutions to participate in this research activity, the current pandemic being one of the motivations, could be avoided. This can be made possible by implementing specific strategies to promote aging research in our country and by exposing the urgent need to establish oral health programs for the benefit of this frail population category. In this respect, the present pilot study may constitute an important starting point to achieve greater willingness to participate in further studies.

Even so, due the scarcity of similarly reports on this topic in Romania, our research highlights the need for and importance of adequate knowledge of the oral health and prosthetic status of institutionalized elderly people, given the probable impact on their OHRQoL. The perception of the life quality is based on both sociodemographic indicators (with great global variability) and local ones. The recognition of these factors is mandatory in order to improve the social importance of the oral health of the institutionalized elderly, and to provide good oral prophylaxis and treatment programs, especially in the rural regions of our country. These findings may help to empowering this elderly community with information, and encourage public health and political decision-makers to prioritize the actions needed to direct more resources into supporting the oral health of the institutionalized elderly.

## 5. Conclusions

The elderly residents of care homes in the vicinity of Bucharest had poor oral health, the prevalence of untreated edentulism being critically high, thus highlighting the need for early prosthodontic procedures in this population segment of our society. 

Complete edentation, the most common form of edentation among institutionalized geriatrics, increases with age. The correlation between complete mandibular prostheses and the quality of life of edentulous patients is the strongest. Mandibular edentation predominates in the youngest-old elderly, while maxillary complete edentation is most prevalent in the middle-old, and oldest-old subgroups. Although the prevalence of edentulism in our geriatric sample was paradoxically high and the use of dentures low, the general OHRQoL perception was not unsatisfactory, as indicated by the mean score of OHIP-14. As a trend, the most affected by the impact of oral health on the quality of life, had been female, subjects from rural areas, and those with complete edentations attributed to periodontal etiology.

## Figures and Tables

**Table 1 ijerph-18-06663-t001:** Sociodemographic data according to different age groups.

Variable	Youngest-Old (*n*; %)	Middle-Old (*n*; %)	Oldest-Old (*n*; %)	*p*
Male gender	4 (25%)	6 (21.4%)	2 (14.3%)	0.841
Highest completed education level				0.942
Secondary school	5 (31.2%)	11 (39.3%)	4 (28.6%)	
High school	6 (37.5%)	10 (35.7%)	5 (35.7%)	
University	5 (31.2%)	7 (25%)	5 (35.7%)	
From urban areas	6 (37.5%)	16 (57.1%)	9 (64.3%)	0.294

**Table 2 ijerph-18-06663-t002:** Oral and prosthetic status in elderly institutionalized according to different age groups.

Variable	Youngest-Old (*n*; %)	Middle-Old (*n*; %)	Oldest-Old (*n*; %)	*p*
Oral Status				
**Complete bimaxillary edentation**	1 (6.2%)	10 (35.7%)	9 (64.3%)	0.005 **
Mandibulary complete edentation; dentate maxilla	3 (18.8%)	3 (10.7%)	0 (0%)	
Maxillary complete edentation; dentate mandible	1 (6.2%)	4 (14.3%)	3 (21.4%)	
Bimaxillary dentate	11 (68.8%)	11 (39.3%)	2 (14.3%)	
**Complete maxillary edentation**	2 (12.5%)	14 (50%)	12 (85.7%)	<0.00 ***
Subtotal edentation	2 (12.5%)	6 (21.4%)	0 (0%)	
Kennedy class I edentation	3 (18.8%)	3 (10.7%)	2 (14.3%)	
Kennedy class II edentation	1 (6.2%)	3 (10.7%)	0 (0%)	
Kennedy class III edentation	8 (50%)	2 (7.1%)	0 (0%)	
**Complete mandibular edentation**	4 (25%)	13 (46.4%)	9 (64.3%)	<0.00 ***
Subtotal edentation	0 (0%)	4 (14.3%)	1 (7.1%)	
Kennedy class I edentation	1 (6.2%)	4 (14.3%)	2 (14.3%)	
Kennedy class II edentation	1 (6.2%)	6 (21.4%)	2 (14.3%)	
Kennedy class III edentation	10 (62.5%)	1 (3.6%)	0 (0%)	
Total number of edentulous spaces (median)	3.5	3	2	0.060
Number of edentulous spaces with no prosthetic treatment (median)	1.5	0	0.5	0.170
**Prosthetic status**				
MAXILLA with no prosthetic treatment	6 (37.5%)	7 (25%)	3 (21.4%)	0.022 *
Complete dentures	1 (6.2%)	11 (39.3%)	9 (64.3%)	
Removable partial dentures	4 (25%)	7 (25%)	2 (14.3%)	
Fixed partial dentures	5 (31.2%)	3 (10.7%)	0 (0%)	
MANDIBLE with no prosthetic treatment	5 (31.2%)	8 (28.6%)	6 (31.6%)	0.286
Complete dentures	3 (18.8%)	10 (35.7%)	5 (35.7%)	
Removable partial dentures	1 (6.2%)	6 (21.4%)	1 (7.1%)	
Fixed partial dentures	7 (43.8%)	4 (14.3%)	2 (14.3%)	

* *p* < 0.05; ** *p* < 0.01; *** *p* < 0.001.

**Table 3 ijerph-18-06663-t003:** OHRQoL related to OHIP-14 scores in the elderly institutionalized sample.

Variable	Median OHIP Scope	*p*
Male:Female	11.5:15.5	0.437
Youngest-olds:Middle-olds: Oldest-olds	14.5:17:12.5	0.836
Cause of tooth loss-caries:periodontal disease:caries and periodontal disease: caries and bruxism	14.5:27:14:6.5	0.350
Highest completed education level-secondary school:highschool: university	14:10:18	0.522
Origin-urban:rural	13:17	0.702
**Edentulism**		
Maxillary complete edentulous:partially edentulous arch	18:13	0.549
Mandibular complete edentulous:partially edentulous arch	18.5:12.5	0.067
Dentate in both jaws:Completely edentulous in the maxilla: Completely edentulous in the mandible:Completely edentulous in both jaws	12.5:13.5:25:18	0.273
**Prosthetic rehabilitation**		
Maxilla-no prosthetic rehabilitation:complete denture rehabilitation:removable partial denture rehabilitation:fixed partial dentures rehabilitation	19:16:12:8.5	0.224
Mandible-no prosthetic rehabilitation:complete removable rehabilitation:partial removable denture rehabilitation:fixed partial dentures rehabilitation	17:19:8.5:13	0.098

## Data Availability

The dataset used and analyzed during the current study is available from the corresponding authors upon request.

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
