# Peer review of "Oral Health Related Quality of Life and Prosthetic Status among Institutionalized Elderly from the Bucharest Area: A Pilot Study"

_ijerph, 2021, doi:10.3390/ijerph18126663_

Round 1

Reviewer 1 Report

The manuscript submitted by Laura Iosif et al described a cross-sectional study of the oral health status of elderly subjects in the Bucharest area. The elderly subjects were classified into three groups depending on their ages, and the relationships among their oral health-related quality of life (OHRQoL) and various subjects’ parameters were statistically accessed. The manuscript itself was well constructed and the presented topic was quite important.

The problem is the small number of subjects enrolled as described in the limitation. Therefore, the authors should carefully discuss the generalization of the observed results. For example, difference categorization, e.g. the use of two groups instead of three groups used in this study, and is one of the ways to confirm the observed trends.

Various statistical tests were used. For all P-values, the statistical tests used were described. Also, not only single asterisk, double and triple asterisks should be used for small p-values and these should define below table 2.

For tables 1 and 2, (n; %) should be moved to the header.

The authors analyzed the difference in several sociodemographic data and oral conditions on three age-dependent groups, including 65-74 years, 75-84 years, and 85 or more years.

As described at the end of the Introduction, the thresholds of these ages were proposed by the Canadian symposium of Geriatrics. However, this study included the small subjects and the numbers of the subjects of these three groups were biased, i.e. n=16, 28, and 14. The authors should use the different thresholds so that the three groups mostly include equals subjects.

In table 3, three a few trends showing monotonous increasing or decreasing according to the age. For example, the median OHIP score of the youngest, middle, and oldest olds were 14.5, 17, and 12.5. Middle groups showed the highest value. This might be caused by the bias of different sample numbers and the grouping condition is not suitable. Of course, the current analysis is one of the statistical results but the analysis using other grouping condition is also important to obtain the stronger conclusion which can be generalized.

As another analytical way, the authors should analyze the individual subject without classifications. For example, hierarchical clustering analysis is one of the analytical approaches. If the clusters were found, the subjects included in a cluster show similar parameter patterns. These unsupervised analyses, instead of conventional multivariate statistical analyses, would help to understand the pattern of the multiple variables.

Author Response

We would like to send you in the attachmment bellow the revised manuscript, according to the suggestions of our reviewers. At the same time, you will find attached the cover leters to them, in which we address punctually the recommendations and suggestions made.

Reviewer 2 Report

The paper reads great and provides a detailed pilot analysis regarding the OHQoL in the elderly. There are a few grammatical errors here and there, which need to be resolved. The paper needs to proofread again. I do not have any comments regarding the methodology and significance of this important issue covered in this paper. Thanks.

Author Response

We would like to send you in the attachmment bellow the revised form of our manuscript, according to the suggestions of our reviewers. At the same time, you will find attached the cover leters to them, in which we address punctually the recommendations and suggestions made.

Reviewer 3 Report

This is a small study based on a convenience sample and reporting descriptive results only. The sample analysed is insufficient (n=58) to do stratified analysis by age groups (youngest-old, middle-old and oldest-old). Table 1 shows that there were <20 participants in two of those groups. These issues preclude any meaningful statistical inference. Most comparisons will not be significant because sample size is limited. What is more, findings cannot be inferred to a population because of the convenience sampling. What is the theoretical population, from which participants were drawn, and to which the findings should be generalised to?

Some of the inclusion and exclusion criteria seems odd. Why selection participants with at least one edentulous space? Or excluding those with mixed rehabilitation (RPD and CRD)? This certainly limited the sample size and the representativeness of the sample.

Was the dentist trained for the examinations? Before or during the survey? What was the level of inter- and intra-examiner reliability (Kappa)? How was the type of prosthesis determined? Clinically or through self-reports? Furthermore, was the reliability of the OHIP-14 tested in the sample? Repeated administration of the instrument among some participants? What was the Cronbach alpha?

Consider reducing the length of the discussion. Several paragraphs are irrelevant to explain the findings. The first five paragraphs could be dropped without affecting the flow of the text. Also, reduce the length of the conclusion, which should only report the main findings of the study. Some of the text in this section is about implications, which belongs to the discussion. Explain in 1-2 sentences what is the main take home message of the study. In that same line, delete the last sentence in the Abstract’s conclusion. This is a recommendation, not a finding of the study.

Author Response

We would like to send you in the attachmment bellow the revised manuscript, according to the suggestions. At the same time, you will find attached the cover leters to them, in which we address punctually the recommendations and suggestions made.
